# HLA-I and HLA-II Peptidomes of SARS-CoV-2: A Review

**DOI:** 10.3390/vaccines11030548

**Published:** 2023-02-25

**Authors:** Nawal Abd El-Baky, Amro A. Amara, Elrashdy M. Redwan

**Affiliations:** 1Protein Research Department, Genetic Engineering and Biotechnology Research Institute (GEBRI), City of Scientific Research and Technological Applications (SRTA-City), New Borg El-Arab City, Alexandria P.O. Box 21934, Egypt; 2Biological Sciences Department, Faculty of Science, King Abdulaziz University, Jeddah P.O. Box 80203, Saudi Arabia

**Keywords:** HLA-I, HLA-II, SARS-CoV-2, immunopeptidomics, MHCs, T-cell-mediated immunity

## Abstract

The adaptive (T-cell-mediated) immune response is a key player in determining the clinical outcome, in addition to neutralizing antibodies, after SARS-CoV-2 infection, as well as supporting the efficacy of vaccines. T cells recognize viral-derived peptides bound to major histocompatibility complexes (MHCs) so that they initiate cell-mediated immunity against SARS-CoV-2 infection or can support developing a high-affinity antibody response. SARS-CoV-2-derived peptides bound to MHCs are characterized via bioinformatics or mass spectrometry on the whole proteome scale, named immunopeptidomics. They can identify potential vaccine targets or therapeutic approaches for SARS-CoV-2 or else may reveal the heterogeneity of clinical outcomes. SARS-CoV-2 epitopes that are naturally processed and presented on the human leukocyte antigen class I (HLA-I) and class II (HLA-II) were identified for immunopeptidomics. Most of the identified SARS-CoV-2 epitopes were canonical and out-of-frame peptides derived from spike and nucleocapsid proteins, followed by membrane proteins, whereby many of which are not caught by existing vaccines and could elicit effective responses of T cells in vivo. This review addresses the detection of SARS-CoV-2 viral epitopes on HLA-I and HLA-II using bioinformatics prediction and mass spectrometry (HLA peptidomics). Profiling the HLA-I and HLA-II peptidomes of SARS-CoV-2 is also detailed.

## 1. Introduction

So far, the ongoing COVID-19 infection has caused more than 650 million confirmed cases and above 6.6 million deaths worldwide, reported by WHO [1]. COVID-19 has directly disturbed the daily life of people across the world and contracted the global economy. With the persistent threat of the virus, boosting immunity via vaccines can provide global protection. Effective protective vaccines, which target the SARS-CoV-2 spike protein, have been developed and approved [2,3,4,5,6]. While most of the efforts toward developing a treatment for SARS-CoV-2 have focused on creating antibody responses against the virus, T-cell-mediated immune responses have played a crucial role in immunity during COVID-19 [7,8]. Contrary to the reduction in stable and specific antibody immunity against viral nucleocapsid and spike proteins, efficient T-cell-mediated immune responses continue to be powerful up to six months post-COVID-19 infection [9].

Mild infection with SARS-CoV-2 is associated with the momentary corresponding activation and contraction of SARS-CoV-2-specific protective CD4^+^ and CD8^+^ T cells, B cells, and antibodies [10,11]. On the other hand, severe infection is associated with high titers of virus-specific antibodies, poorly regulated cytokine and ligand responses, and hyperactivation of innate as well as adaptive immune cells [10,11].

Vaccines designed based on antibody responses may fail to provide protection against antigenically different viral strains (virus variants) [12,13]. T-cell-mediated immune responses involve SARS-CoV-2-specific CD8^+^ cells (cytolytic T cells) that have the ability to lyse infected cells expressing MHC class I (MHC-I) molecules presenting viral peptides and CD4^+^ cells that recognize viral peptides bound to MHC class II (MHC-II) molecules, and kill infected cells or participate in immune response coordination. In contrast to neutralizing antibodies, T cells target viral peptide-MHC epitopes derived from more highly conserved internal viral proteins across antigenically different viral strains, providing universal immunity across unrelated strains [14]. Therefore, vaccines triggering cross-strain protective T cell immunity represent an efficient way to protect against the threat of evolving SARS-CoV-2 variants.

When the virus infects host cells, its antigens are proteolytically processed into peptides and presented on the surface of infected cells by human MHC, known as HLA molecules, to be presented to T cells [15]. This peptidome of HLA represents an immunological signature, which can be particularly recognized by circulating cytotoxic T cells (CD4+ and CD8+ T cells) via their T cell receptor (TCR), resulting in the clearance of infected host cells via the lysis of these cells in addition to catalyzing further immune responses [16]. Indigenous populations express distinctive and unique HLA profiles that vary from other ethnic groups’ profiles [17]. Exploring the SARS-CoV-2-derived HLA peptide repertoire allows for the characterization of viral epitopes, which activate cytotoxic T cells.

B cells recognize external viral epitopes, while the responses of T cells may be elicited by all viral proteins (several canonical and non-canonical (out-of-frame) viral open reading frames (ORFs) in spike and nucleocapsid proteins) [18]. Bioinformatics prediction for the binding affinity of HLA-I and HLA-II was employed to detect the epitopes of SARS-CoV-2, which are recognized by T cells and promote sustainable memory populations [19,20,21,22,23,24,25,26]. The data of identified SARSCoV-2 peptides from bioinformatics prediction are based on the reactivity assays and biochemical binding assays (IMMUNITRACK website, https://www.immunitrack.com, last accessed on 26 February 2022) of predicted peptides [20,25,26,27,28,29].

Though bioinformatics prediction for HLA-I and HLA-II binding is a very convenient tool to detect putative antigens, it has some drawbacks. First, all predicted SARS-CoV-2 peptides’ processing and presentation are not certainly performed by infected cells or antigen-presenting cells (APCs) during infection. Second, viral antigen processing and presentation are complex biological pathways with multiple steps [30], and only some of these steps are justified by current computational predictors, achieving an average positive predictive value across HLA alleles of 64% [31]. Third, models of bioinformatics prediction do not justify the manipulation ways of viruses to cellular processes, which affect antigen presentation [32,33]. Fourth, models of prediction do not capture the viral antigen expression and epitope presentation during the course of infection; HLA-I presentation of viral epitopes can peak post-infection [34,35]. To overcome these limitations, bioinformatics tools should be combined with experimental measurements of naturally presented viral peptides upon infection (HLA peptidomics) to deepen our understanding of T-cell-mediated immune responses to SARS-CoV-2 [36]. Recent studies have successfully identified presented HLA-I and HLA-II SARS-CoV-2 antigens via HLA peptidomics [18,36,37,38]. 

This review addresses cell-mediated immune responses against SARS-CoV-2 immunopeptidomics to study and fight SARS-CoV-2 infection, as well as the identification of presented HLA-I and HLA-II SARS-CoV-2 antigens via bioinformatics prediction and HLA peptidomics.

## 2. Cellular Immune Responses against SARS-CoV-2

T-cell-mediated immune responses play a crucial role in SARS-CoV-2 control and their significance may have been quite underrated so far. A subgroup of T cells primed against severe acute respiratory syndrome (SARS), Middle East respiratory syndrome (MERS), and other coronaviruses can cross-react with SARS-CoV-2 and might participate in clinical protection, mostly in early life [39], and the references within]. The memory of T cells has sustained a breadth of recognition of viral proteins, valued at about 30 epitopes within each individual, which can limit viral mutations’ impact within an individual and support protection against viral variants, comprising Omicron [39]. Current vaccines for COVID-19 can prompt robust and persistent T-cell-mediated immunity resulting in noteworthy protection against hospitalization or death. The further enhancement of cellular immune responses can be achieved by innovative or heterologous regimens. Yan et al. (2022) reported that the natural infection of SARS-CoV-2 stimulated a robust and sustained memory T cell immune response as well as neutralizing antibodies in most COVID-19 patients, which may significantly contribute to protection against reinfection [40].

The course of COVID-19 infection can be divided into three stages, based on former research on seasonal coronaviruses besides clinical remarks in COVID-19 patients [41,42]. The first stage of the COVID-19 infection course includes the period from 0 days to 7–10 days, in which infected patients develop no symptoms or develop mild to moderate influenza-like symptoms and the detection of the virus can be performed via polymerase chain reaction (PCR) analysis [41,43,44]. The second and third stages include the period from 7–10 to 14–21 days, and the period from 14–21 days or more after the onset of symptoms, respectively. During the second and third stages, if patients have an effective immune function, they will enter the recovery phase and the virus can be suppressed. Yet, patients with dysfunctional immunity depending on gender, age, or other unidentified factors will progress to a severe phase and the virus cannot be suppressed efficiently. In other words, patients with a mild infection will eventually recover after entering the second stage, while patients with non-mild infection worsen and present dyspnea and/or severe hypoxemia that quickly progress to acute respiratory distress syndrome (ARDS) after approximately 8 days after the onset of symptoms [45]. Severe conditions in the second stage include ARDS, coagulation dysfunction, difficulty in correcting metabolic acidosis, septic shock, and multiple organ failure [46]. Some patients with severe infection survive and others cannot overcome the infection and ultimately decease.

The immune system of infected patients responds to SARS-CoV-2 in three stages (Figure 1). The severity of infection is predominantly related to the cell-mediated immune responses of the host. Patients with mild infection and those with severe but improved infection can display a normal immune response to efficiently eradicate the virus. The immune systems of patients with fatal severe infection respond to SARS-CoV-2 in three stages: normal or hypofunction, hyperactivation, and anergy [47], while the immune responses of mild patients and severe survived patients are restricted to the stages of normal or hypofunction and hyperactivation.

Patients with mild COVID-19 infection in the normal or hypofunction immune response stage have a white blood cell (WBC) count that is more decreased than normal [46]. The count of lymphocytes in these patients is lower than in healthy individuals but is still higher than in patients with severe infection [48]. Additionally, T cell function in mild patients is higher than that in severe patients [48]. 

On the other hand, severe patients in the normal or hypofunction immune response stage exhibit a significant decrease in CD4^+^ and CD8^+^ T cell count and function, especially in deceased patients compared to survived ones [49]. Furthermore, severe patients in this stage display an increase in neutrophil (N) count [49,50], a significant decrease in dendritic cell (DC) count in deceased patients compared to survivors [51], a reduced count and impaired function of plasmacytoid dendritic cells (pDCs) [52], and a decrease in natural killer (NK) cell count in early infection, but severe patients have a higher NK cell count than mild patients [48].

In the hyperactivation immune response stage, mild patients recover as their T cell and B cell counts gradually increase [48,49]. The count of their plasma cells as well as the activity of their CD4^+^ and CD8^+^ T cells also increase, but their CD8^+^ T cell count decreases. In contrast, severe patients exhibit further decreases in the count of T cells and B cells [53,54]. The overactivation of CD4^+^ and CD8^+^ T cells appears in deceased patients at this stage more than in those recovering from severe illness [55,56]. Deceased patients display a decrease in the count and activity of monocytes, DCs, and NK cells, while survived patients exhibit an increase in monocyte count and activity [53,57]. Moreover, emergency myelopoiesis (release of immature neutrophils and increase in mature/partially activated neutrophils) occurs in severe patients [49].

Following the above-mentioned first and second stages of the immune response of COVID-19 patients to the virus, if patients have an effective immune function, the virus can be effectively suppressed. Consequently, patients can enter the recovery period. On the contrary, patients with the impaired immune function will enter a state of incompetence of their immune system (anergy) and inability to resist the infection, and ultimately decease. The numbers of T lymphocytes, B lymphocytes, monocytes, DCs, and NK cells continue to decrease in patients with fatal severe infection [48].

Based on these immune response kinetics during COVID-19 infection, type I interferon treatment was suggested for patients with a severe infection in the hypofunctional stage, in addition to low molecular weight heparin (LMWH), glucocorticoid therapy, intravenous immunoglobulin (IVIG), and antibiotics in the hyperactivation stage [47].

## 3. Immunopeptidomics to Study and Fight SARS-CoV-2

The recognition of T cells to viral-derived peptides bound to MHCs (HLA molecules) is essential to initiate cell-mediated immunity against viral infection or support developing high-affinity antibody responses [58]. Thus, detecting these viral antigens presented on HLA molecules and recognized by T cells is vital for the elucidation of natural immune responses to viral infection and developing efficient new vaccines or effective approaches for treating COVID-19 [59]. 

Remarkably, currently developed COVID-19 vaccines are targeting the stimulation of neutralizing antibodies to the virus via immunization with the viral spike protein or only the receptor binding domain (RBD) of the spike protein [60]. Studies have shown that recovered individuals from SARS-CoV (the closely related coronavirus to SARS-CoV-2) have persistent long-lived memory CD8^+^ T cell responses against viral nucleocapsid for a period from six to eleven years, while memory B cell responses and antiviral antibodies are lacking after six years [61,62]. Likewise, most COVID-19 patients have detectable antibody responses to the virus ten to fifteen days after the onset of symptoms, but these responses decline to the baseline within three months in many patients [63]. Therefore, vaccines that target stimulating neutralizing antibodies to the SARS-CoV-2 spike protein cannot provide long-term immunity.

In SARS-CoV-infected mice, responses of CD8^+^ T cells are sufficient to clear the virus and protect them from the clinical disease [64]. Interestingly, the immunization of mice with a single immunodominant epitope of virus-specific memory CD8^+^ T cells could offer substantial protection against lethal SARS-CoV infection [65]. Consequently, understanding the natural immune responses of CD8^+^ T cells to SARS-CoV-2 has a significant impact on designing more durable COVID-19 vaccines. However, understanding the natural immune responses of human CD8+ T cells is complicated since these cells are restricted by hyperpolymorphic HLA proteins, which require the identification of T cell epitopes for different HLA alleles across ethnically varied global populations [17]. This wide-ranging HLA polymorphism makes the identification of viral epitopes a considerable challenge. There are more than 35,000 alleles of human HLA that have been identified in the IPD-IMGT/HLA Database [66]. Among these, more than 25,000 HLA-I and 9500 HLA-II alleles are currently included in the database. In silico and experimental approaches were employed to deal with the large diversity of HLA alleles and the peptide arrays that they can present (Figure 2).

As an example, immunopeptidomics was employed to detect the sequence of HLA-bound peptides presented on cells infected with influenza and other respiratory viral infections via liquid chromatography and tandem mass spectrometry [17]. Immunopeptidomics could successfully and comprehensively identify the epitopes of influenza-specific CD8^+^ T cells restricted by the HLA alleles of Indigenous Australians using antigen-presenting cell lines expressing Indigenous Australians’ HLA alleles [17]. As of February 2022, above 970,000 MHC ligands identified using mass spectrometry were collected within the Immune Epitope Database (IEDB) [67]. Among these ligands, more than 800 ligands (non-redundant by either sequence or modification) have derived from SARS-CoV-2.

Bioinformatics prediction and mass spectrometry were used to determine the SARS-CoV-2 peptide sequences in HLA-I and HLA-II peptidomes that are specifically recognized by circulating the CD8^+^ and CD4^+^ T cells of COVID-19 patients [18,36,37,38,59]. These peptides were found to be derived from canonical and out-of-frame ORFs in SARS-CoV-2 nucleocapsid and spike proteins, which are not caught by existing vaccines and stimulated effective responses of T cells in a mouse model along with COVID-19 patients.

Viral epitopes were identified for each of the six most prevalent HLA-I alleles across COVID-19 patients [59]. These epitopes were found in virus regions that were not subjected to mutational variation. Remarkably, just three out of twenty-nine shared CD8^+^ T cell epitopes were found in the SARS-CoV-2 spike protein, while most epitopes were positioned in the nucleocapsid protein or the ORF1ab polyprotein [59]. In general, CD8^+^ T cells did not cross-react with epitopes in the four seasonal coronavirus strains that give rise to common cold symptoms. These data may inform innovative vaccine development and help to achieve enhanced natural immunity of CD8^+^ T cells against SARS-CoV-2.

Li et al. (2021) analyzed a library of about 40,000 synthetic peptides via tandem mass spectrometry and proteomic procedures to create a large dataset of the MS/MS spectra of peptides derived from SARS-CoV-2 [68]. They constructed an online knowledgebase named virus MS (https://virusms.erc.monash.edu/, last accessed on 18 January 2023) for detailed documentation and annotation, besides the analysis of these synthetic peptides, comprising experimental data, peptide modifications, predicted binding affinities of HLA for these peptides, and their MS/MS spectral information. 

A more recent study reported that the changes in host cells in response to SARS-CoV-2 infection were comprehensively interpreted by integrated immunopeptidomics and proteomics [69]. This study revealed that the initial innate immune response of Calu-3 cells was via TLR3, and then interferon signaling pathway activation. Additionally, host cells were found to present SARS-CoV-2 antigens on both HLA-I and HLA-II for recognition by adaptive immune cells. MHC immunoprecipitation and glycosylation analysis indicated that SARS-CoV-2 could significantly disrupt antigen presentation by causing a higher level of HLA proteins and affecting both the synthesis and degradation of HLA proteins. This work sheds light on the host response to COVID-19 infection and can aid in developing therapeutics and vaccines for the pandemic.

## 4. Immunopeptidomic Screenings for SARS-CoV-2 Using Bioinformatics

Various studies have predicted SARS-CoV-2 epitopes across different HLA alleles utilizing overlapping peptide tiling and/or bioinformatics [19,21,26,59,70,71,72]. Identified T-cell epitopes in COVID-19 patients via bioinformatics are summarized in Table 1. Campbell et al. (2020) predicted SARS-CoV-2 epitopes across 9360 HLA-I alleles (HLA-A, -B, and -C) [19]. They created a database of epitopes predicted to bind any HLA-I protein across the entire proteome of SARS-CoV-2, which can be accessed publicly. They found 6748 unique combinations of viral peptides (derived from all 11 proteins spanning the entire viral peptidome) and HLA alleles, which have a predicted <500 nM binding affinity, comprising 1103 unique peptides and 1022 HLA alleles [19].

Ferretti et al. (2020) generated a library of protein fragments with a length of 61 amino acids, which tiled across all 11 SARS-CoV-2 ORFs in steps of 20 amino acids [59]. They included all protein-coding genetic variants from the 104 SARSCoV-2 isolates reported as of 15 March 2020. Each protein fragment was represented 10 times, each encoded with a distinctive barcode of nucleotide to offer internal replicates in their screens, to achieve 43,420 clones of library size [59]. They identified 29 shared CD8^+^ T cell epitopes in COVID-19 convalescent patients across six HLA alleles: A*02:01, A*01:01, A*03:01, A*11:01, A*24:02, and B*07:02, with an affinity (equilibrium dissociation constant) of 2.8–206 nM as predicted by NetMHC4.0 [59,73,74].

Nguyen et al. (2020) performed an extensive in silico analysis of the binding affinity between all SARS-CoV-2 peptides and 145 HLA-I genotypes (HLA-A, -B, and -C) [70]. They reported that HLA-B*46:01 had the fewest predicted binding peptides for SARS-CoV-2, while HLA-B*15:03 had the greatest capacity to present highly conserved SARS-CoV-2 peptides, which are shared among the other four seasonal coronaviruses. These findings suggest that individuals with the HLA-B*46:01 allele may be more susceptible to COVID-19, while HLA-B*15:03 could enable cross-protective T-cell-mediated immune responses.

Li et al. (2021) synthesized a peptide library of 1809 peptides, each with a length of fifteen amino acids and overlapping by nine amino acids, from the entire proteome of SARS-CoV-2 [68]. They conducted two main predictions for the binding affinity between peptides and HLA-I alleles, comprising (1) using peptides from tandem mass spectrometry experiments and (2) proteome-wide binding prediction. These two datasets harbored a total of 39,650 synthetic peptides, which were harvested, documented, annotated, and analyzed in the online knowledgebase virusMS constructed by the authors [68]. They employed NetMHC4.0 to predict the binding affinity between peptides and 12 HLA-I alleles including A*01:01, A*02:01, A*03:01, A*24:02, A*26:01, B*07:02, B*08:01, B*27:05, B*39:01, B*40:01, B*58:01, and B*15:01 [68].

Saini et al. (2021) experimentally evaluated 3141 HLA-I-binding peptides covering the complete genome of SARS-CoV-2 utilizing DNA-barcoded peptide-MHC complex (pMHC) multimers combined with a T cell phenotype panel to detect immunogenic epitopes recognized by CD8+ T cells [71]. They first selected 2204 potential HLA-I-binding peptides with a length of 9–11 amino acids using NetMHCpan 4.1 for experimental evaluation. The selected peptides were predicted to bind one or more of ten prevalent HLA-I alleles (A*01:01, A*02:01, A*03:01, A*24:02, B*07:02, B*08:01, B*15:01, C*06:02, C*07:01, and C*07:02), leading to a total of 3141 HLA-I-binding peptides for experimental evaluation [71]. They reported an extensive list of 122 immunogenic and a subclass of immunodominant T-cell epitopes of SARS-CoV-2. Immunogenic regions were mostly derived from viral ORF1 and ORF3, and the immunodominant epitopes were mainly found in ORF1.

Tarke et al. (2021) studied CD8+ and CD4+ T cell epitopes in 99 COVID-19 convalescent cases [72]. The entire proteome of SARS-CoV-2 was probed using 1925 peptides, to ensure an unbiased coverage of HLA-II allele responses. They also studied 5600 predicted binding epitopes for 28 HLA-I alleles. This work could identify several hundreds of SARS-CoV-2-derived epitopes restricted by HLA.

## 5. Immunopeptidomic Screenings for SARS-CoV-2 Using Mass Spectrometry

Immunopeptidomics for the investigation of peptides presented by HLA proteins using mass spectrometry were previously used in designing vaccines against human herpes virus-6B [75] and tuberculosis [76], but have been highlighted during the COVID-19 pandemic. As the scientific community has started protective vaccine development against the SARS-CoV-2 spike glycoprotein, a novel immunopeptidomics investigation was started to detect the peptides of the SARS-CoV-2 spike presented on HLA-II when dendritic cells from healthy donors were pulsed with the viral spike [38].

Vaccine development against COVID-19 is mainly based on humoral responses targeted at the SARS-CoV-2 spike because this protein drives the viral interactions of cellular binding and entry. The RBD of the spike protein interacts with the angiotensin-converting enzyme 2 (ACE2) in humans to enable cellular entry [77]. Thus, some early models during the initial stages of developing COVID-19 vaccines targeted the SARS-CoV-2 spike RBD [78].

Using the MHC-associated peptide proteomics (MAPPs) method, human DCs were intentionally pulsed with a SARS-CoV-2 spike [38]. Scientists isolated CD14^+^ monocytes from healthy human donors (blood samples were collected before the occurrence of the COVID-19 pandemic) to certify that there was no previous immune knowledge of the virus and then differentiated them into immature DCs. Immature DCs were treated with viral spike glycoprotein. HLA-II molecules were isolated from the mature and lysed cells using immunoprecipitation and HLA-II antigen-derived peptides were detected using highly sensitive liquid chromatography-tandem mass spectrometry (LC-MS/MS). Mass spectrometry could identify 876 peptide sequences from the SARS-CoV-2 spike, whereby 526 sequences of which were found to be unique [38]. Dendritic cells could present peptides, which span the entire viral spike protein. Identified HLA-II peptides from 11 regions were presented by the most analyzed donors. The correlation between presented and predicted peptides was found to be low when the results of this experiment were compared to the predictive algorithms [38]. Only two of the eleven observed HLA-II peptides of the SARS-CoV-2 spike protein through immunopeptidomics were predicted by the predictive algorithm (the TepiTool resource in IEDB) for dominant HLA-II peptides published by Grifoni et al. (2020) [21].

### 5.1. Profiling HLA-I Peptidome of SARS-CoV-2

To analyze the HLA-I peptidome of SARS-CoV-2 infection, HLA-I proteins were immunoprecipitated from human embryonic kidney HEK293T cells and human lung A549 cells infected with SARS-CoV-2 [18]. These cell lines were chosen based on biological relevance and high coverage of HLA-I alleles, and had been previously transduced with ACE2 and TMPRSS2, two known factors of viral entry. Their HLA-I-bound peptides were analyzed via LC-MS/MS. Additionally, the authors analyzed the entire proteome of the flowthrough of immunoprecipitation via LC-MS/MS and examined the SARS-CoV-2 effect on the expression of a human gene using RNA-seq [18]. This study reported the detection of 28 peptides derived from SARS-CoV-2 canonical proteins (non-structural proteins nsp1-nsp3, nsp5, nsp8, nsp10, nsp14, and nsp15, nucleocapsid, membrane, spike, and ORF7a) (Table 2, Figure 3A). Surprisingly, 25% (nine peptides) of detected HLA-I peptides were derived from out-of-frame ORFs in viral spike and nucleocapsid proteins (Table 2, Figure 3A). Some of these out-of-frame HLA-I peptides could elicit more potent T cell responses (measured using an IFN-γ ELISpot assay) in transgenic HLA-A2 mice and COVID-19 peripheral blood mononuclear cells (PBMCs) from COVID-19 convalescent patients than canonical peptides [18].

Unexpectedly, lower nucleocapsid representation (single HLA-I peptide from nucleocapsid) was observed [18], although previous studies of RNA-seq and ribosome profiling (Ribo-seq) have revealed a high abundance of this SARS-CoV-2 protein [79,80]. The authors experimentally correlated this lower representation of the nucleocapsid protein to the fact that this protein can harbor fewer peptides compatible with the binding motifs of HLA.

The non-canonical peptide from ORF9b, ELPDEFVVVTV, was found to be in the top five reactive peptides [18]. This peptide stimulated the strongest response of CD8^+^ cells among all tested HLA-I peptides. Inspecting the profile of gene expression and the sequence of TCR of the reacting T cells provided supporting evidence for the ELPDEFVVVTV epitope functional relevance during the course of COVID-19. Most ELPDEFVVVTV-reactive T cells exhibited a high expression of effector and memory markers based on gene sets described in the profiling study of the COVID-19 CD8^+^ subpopulation by Su et al. (2020) [81].

Nagler et al. (2021) identified the SARS-CoV-2 HLA-I peptides presented by a wide range of the human population, using mono-allelic 721.221 B cells that express the most frequent HLA-I alleles or multi-allelic IHW01161 and IHW01070 B cells and the lung adenocarcinoma cell line Calu-6 which expresses the most frequent HLA-I alleles endogenously [36]. Both of these cell systems were transduced with viral genes or were infected with SARS-CoV-2, followed by HLA peptidomics. B cells were chosen because they play a key role in presenting the viral peptides to the immune system when infected by viruses [82,83] and Calu-6, due to their relevance in the disease. They identified two HLA-I peptides derived from internal out-of-frame ORFs (ORF S.iORF1/2 and ORF9b) found in the spike and nucleocapsid coding region (Table 2, Figure 3A). Additionally, they detected ten HLA-I peptides derived from canonical SARS-CoV-2 ORFs (two peptides derived from the spike, two from nsp3, one from ORF3a, one from nsp1, and four from nucleocapsid) (Table 2, Figure 3A). The study also reported three shared SARS-CoV-2-derived HLA-I peptides in different cell types [36]. The first peptide NSSPDDQIGYY was derived from nucleocapsid. The second shared peptide was APRITFGGP, which was also derived from nucleocapsid. The third peptide FLLPSLATV was derived from nsp6.

Besides the effect of HLA-I polymorphism on its immunopeptidome, the influence of polymorphism of endoplasmic reticulum aminopeptidases (ERAPs), especially endoplasmic reticulum aminopeptidases 1 and 2 (ERAP1 and ERAP2) on HLA-I antigen processing, has drawn scientific attention as well. These aminopeptidases are responsible for the trimming of antigenic peptides, previously processed in the cytoplasm by the proteasome, within the endoplasmic reticulum, thus conditioning HLA-I antigen presentation [84]. ERAP1 and ERAP2 generate antigenic peptides that have a length of 8–9 amino acids, which efficiently accommodate within the HLA-I binding groove. This conditioning occurs in physiological as well as pathological contexts such as infection mediated by SARS-CoV-2 [85,86]. Stamatakis et al. (2021) addressed the fundamental mechanisms behind immune response breadths between individuals and the highly variable severity of COVID-19 infection via analyzing the proteolytic processing of antigenic peptides of the viral spike using ten common allotypes of ERAP1 [87]. To achieve better emulation for intracellular antigen processing, they employed a systematic proteomic approach to concurrently analyze hundreds of trimming reactions in parallel. They found that all allotypes of ERAP1 could produce optimal ligands for HLA-I, including known epitopes of SARS-CoV-2, but significantly differed in produced peptide sequences, which suggests allotype-dependent sequence biases [87]. Allotype 10 was found to be functionally distinctive from other allotypes. The study reported that common ERAP1 allotypes could cause significant antigen processing heterogeneity and through this mechanism contribute to variable immune responses in COVID-19 infection [87]. D’Amico et al. (2021) reported that the dysfunctional status of ERAP1 and ERAP2 can worsen the effect of SARS-CoV-2 infection on the renin-angiotensin system (RAS) because these two enzymes are involved in RAS regulation and are key components of HLA-I antigen processing, which aggravates the clinical outcome of the disease [88].

### 5.2. Profiling HLA-II Peptidome of SARS-CoV-2 

Nagler et al. (2021) identified overlapping peptides derived from SARS-CoV-2 membrane and nucleocapsid proteins, which were presented on HLA-I as well as HLA-II molecules [36]. The study specifically reported the detection of a nested set of 25 peptides derived from the membrane protein and presented on the HLA-II B1*01:02 allele in 721.221 B cells (Table 3, Figure 3B). Moreover, nine nested peptides derived from nucleocapsid that were presented on the HLA-II B1*01:02 allele were identified in 721.221 cells (Table 3, Figure 3B). The HLA-II-bound peptide KDGIIWVATEGALN and HLA-I-bound peptide ATEGALNTPK derived from the SARS-CoV-2 nucleocapsid exhibited an overlap of seven amino acids and were identified in 721.221 cells presented on the HLA-II A11:01 allele [36]. The HLA-II peptide LSYYKLGASQRVAGD identified in this study was previously found to be recognized by CD4^+^ T cells in twelve COVID-19 patients [25].

Parker et al. (2020) profiled the HLA-II-bound peptide repertoire presented by dendritic cells pulsed with a SARS-CoV-2 spike [37]. They identified 209 unique sequences of HLA-II-bound peptides, many of which were found in nested sets (Table 3, Figure 3B). These peptides map to sites throughout the SARS-CoV-2 spike including glycosylated regions. This study also highlighted the receptor-binding motif (RBM) in the SARS-CoV-2 spike as being a region rich in HLA-DR-binding peptides (Table 3, Figure 3B); combined with the previous studies [15,34,72] on mice immunized with recombinant DNA vectors encoding the S-protein from SARS and SARS-CoV-2, it can be suggested that the peptides created from this region may be presented in a cross-species manner.

Becerra-Artiles et al. (2022) profiled the immunopeptidome of human coronavirus OC43-infected cells, identified epitopes of CD4^+^ T cells specific to seasonal coronaviruses, and found two HLA-II-bound peptides that cross-react with SARS-CoV-2 [89]. These two peptides are RSAIEDLLFDKVKLS and LTALNAYVSQQLSDS, which are derived from the viral spike protein and bind to HLA-DP4 and HLA- DR2b, respectively.

Finally, all of these immunopeptidomics studies on the processing and presentation of viral antigens and the restriction of HLA molecules, along with studies that explore the antigenic landscape to achieve immune targeting at the pre-clinical stage, pave the way toward clinical trials where immunopeptidomics are directly implemented in the conception of innovative vaccines and treatments for SARS-CoV-2 [90,91,92].

## 6. Conclusions

Immunopeptidomics pave the way toward decoding SARS-CoV-2 peptides to deliver more potent therapeutics or vaccines. Deepening our understanding of the peptide-binding motifs displayed by HLA-I and HLA-II proteins is crucial to elucidate the peptides that they are expected to present and, consequently, the T-cell-mediated immunity specificity to certain proteins displayed by SARS-CoV-2. Combining algorithms with the potent separation and detection facilities of liquid chromatography in addition to tandem mass spectrometry makes the mapping of HLA-I- and HLA-II-binding SARS-CoV-2 peptides accurate. Overall, the mapping results revealed that most of the HLA-I- and HLA-II-bound peptides were derived from SARS-CoV-2 spike and nucleocapsid proteins, followed by membrane proteins, whereby many of which are out-of-frame peptides or are not caught by developed vaccines and could elicit effective responses of T cells in vivo. The spike protein, principally its RBM, is a rich region in HLA-II-binding peptides. Approximately a quarter of the mapped HLA-I peptides are derived from out-of-frame ORFs in viral spike and nucleocapsid proteins. Some of these out-of-frame peptides were proved to be more immunogenic than canonical peptides. Early expressed SARS-CoV-2 proteins could dominate HLA-I presentation as well as immunogenicity. These findings enhance the possibilities for developing COVID-19 vaccines and treatments with greater specificity as well as triggering broader immunity at the population level.

## Figures and Tables

**Figure 1 vaccines-11-00548-f001:**
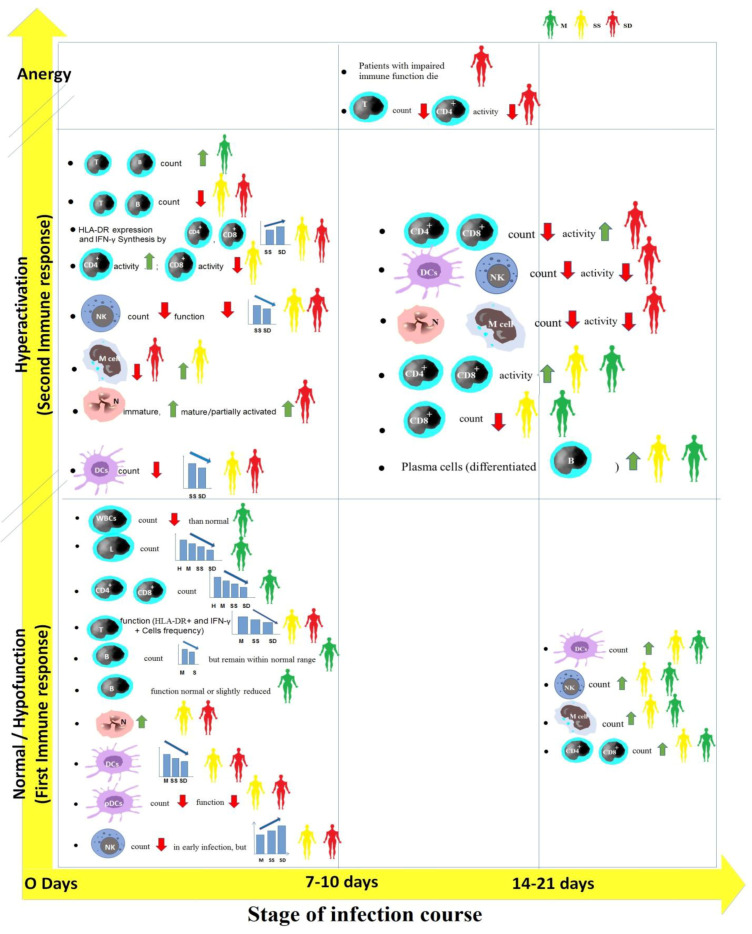
Cell-mediated immune responses against SARS-CoV-2 in COVID-19 patients with different clinical outcomes in the three stages of the infection course. H, healthy individuals; M, mild patients; S, severe patients; SS, severe survived patients; and SD, severe deceased patients. Green up arrows or red down arrows mean that compared with the result in the early stage, the count or activity of immune cells in the defined stage has increased or decreased. WBCs, white blood cells; T, T lymphocytes; B, B lymphocytes; N, neutrophil; DCs, dendritic cells; pDCs, plasmacytoid dendritic cells; NK, natural killer cell; and M cell, monocyte.

**Figure 2 vaccines-11-00548-f002:**
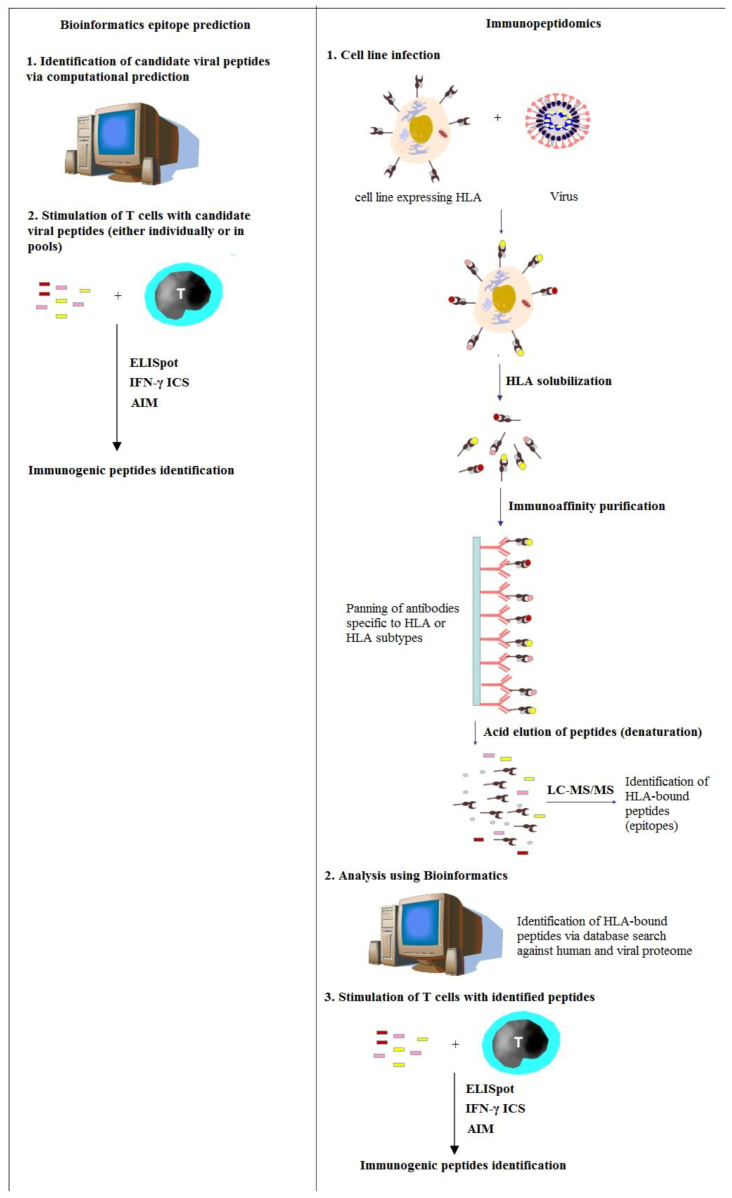
Bioinformatics epitope prediction and immunopeptidomics approach to identifying epitopes of T cells for HLA alleles. ELISpot, the enzyme-linked immunosorbent spot; ICS, intracellular cytokine staining; and AIM, activation-induced markers.

**Figure 3 vaccines-11-00548-f003:**
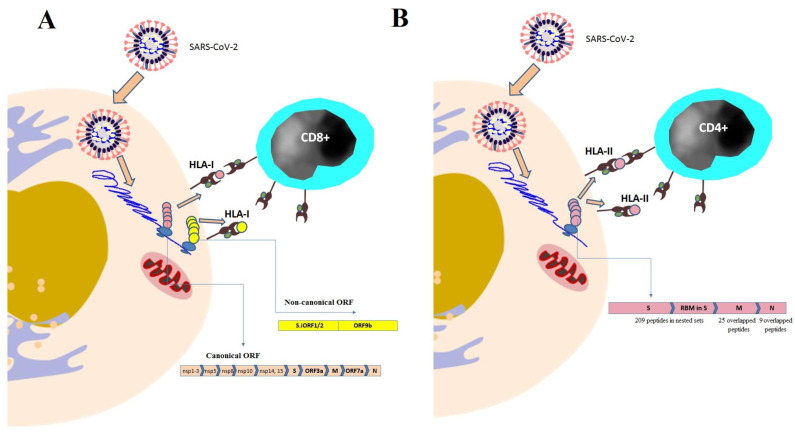
(**A**). Profiling HLA-I peptidome of SARS-CoV-2. nsp, non-structural protein; S, spike protein; M, membrane protein; and N, nucleocapsid protein. (**B**). Profiling HLA-II peptidome of SARS-CoV-2. S, spike protein; RBM, receptor-binding motif in spike; M, membrane protein; and N, nucleocapsid protein.

**Table 1 vaccines-11-00548-t001:** Identified T cell epitopes in COVID-19 patients using bioinformatics.

HLA Allele	Peptide of SARS-CoV-2	Parent Protein	Ref.
HLA-B15:03	PRWYFYYLGTGP WSFNPETN QPPGTGKSH and VYTACSHAAVDALCEKA	Nucleocapsid Membrane ORF1ab	[70]
A01:01 B07:02	TTDPSFLGRY SPRWYFYYL	ORF1 Nucleaocapside	[71]
B1*01:01, B1*03:01 B1*04:01, B1*04:05 B1*07:01, B1*08:02 B1*09:01, B1*11:01 B1*12:01, B1*13:02 B1*15:01, B3*01:01 B3*02:02, B4*01:01 B5*01:01, A1*01:03 A1*02:01, A1*03:01 A1*01:01, A1*01:02 A1*04:01, A1*05:01	1925 peptides	Non-structural Proteins (nsp1- nsp16) Spike ORF3a Envelope Membrane ORF6 ORF7a ORF8 Nucleaocapside ORF10	[72]
A01:01, A02:01 A03:01, A11:01 A24:02, A26:01 A29:02, A30:01 A32:01, A68:01 B07:02, B08:01 B15:01, B35:01 B40:01, B44:02 B44:03, B51:01	5600 predicted binding epitopes	Non-structural Proteins (nsp1- nsp16) Spike ORF3a Envelope Membrane ORF6 ORF7a ORF8 Nucleocapsid ORF10	[72]
1022 HLA-A, -B, and -C	1103 peptides	ORF1ab Spike Nucleocapsid ORF3a Membrane ORF8 ORF7a Envelope ORF6 ORF10 ORF7b	[19]
B1*04:01	IRGWIFGTTLDSKTQSLL CTFEYVSQPFLMD QPFLMDLEGKQGN TRFQTLLALHRSYLTPGD SSSGW KSFTVEKGIYQTSNFRVQ YLYRLFRKSNLKPFERDI KPFERDISTEIYQ QSIIAYTMSLGAENSVAY VKQIYKTPPIKDFGGFNF DSLSSTASALGKLQDVV QLIRAAEIRASANLAATK HWFVTQRNFYEPQII	Spike Spike Spike Spike Spike Spike Spike Spike Spike Spike Spike Spike	[21]
B1*07:01	KSFTVEKGIYQTSNFRVQ SASFSTFKCYGVSPTKL QSIIAYTMSLGAENSVAY	Spike Spike Spike	[21]
A*02:01	KLPDDFTGCV SIIAYTMSL ALNTLVKQL VLNDILSRL LITGRLQSL RLNEVAKNL NLNESLIDL FIAGLIAIV TLACFVLAAV GLMWLSYFI ALNTPKDHI LQLPQGTTL LALLLLDRL LLLDRLNQL RLNQLESKM GMSRIGMEV CLEASFNYL WLMWLIINL ILLLDQALV SACVLAAEC SLPGVFCGV TLMNVLTLV SMWALIISV	Spike Spike Spike Spike Spike Spike Spike Spike Membrane Membrane Nucleocapsid Nucleocapsid Nucleocapsid Nucleocapsid Nucleocapsid Nucleocapsid ORF1ab ORF1ab ORF1ab ORF1ab ORF1ab ORF1ab ORF1ab	[21]
HLA-I	HLRIAGHHL TKAYNVTQAF	Membrane Nucleocapsid	[21]
HLA-II	NLDSKVGGNYNYLYRLFR	Spike	[21]
B58:01	RIFTIGTVTLKQGEI TVTLKQGEI	ORF3a ORF3a	[21]
B40:01	GDAALALLLL MEVTPSGTWL	Nucleocapsid Nucleocapsid	[21]
A24:02	NFKDQVILL	Nucleocapsid	[21]
HLA-A02:01	KLDDKDPNF FGDDTVIEV FLAFVVFL KLNDLCFTNV FLFLTWICL	Nucleocapsid ORF1ab Envelope Spike Membrane	[26]
A02:01	KLWAQCVQL YLFDESGEFKL LLYDANYFL ALWEIQQVV YLQPRTFLL LLLDRLNQL	ORF1ab ORF1ab ORF3a ORF1ab Spike Nucleocapsid	[59]
A01:01	GTDLEGNFY NTCDGTTFTY CTDDNALAYY TTDPSFLGRY PTDNYITTY FTSDYYQLY ATSRTLSYY DTDFVNEFY	ORF1ab ORF1ab ORF1ab ORF1ab ORF1ab ORF3a Membrane ORF1ab	[59]
A03:01	KTIQPRVEK VTNNTFTLK KCYGVSPTK KTFPPTEPK	ORF1ab ORF1ab Spike Nucleocapsid	[59]
A11:01	VTDTPKGPK ATSRTLSYYK ASAFFGMSR ATEGALNTPK KTF KTFPPTEPK	ORF1ab Membrane Nucleocapsid Nucleocapsid Nucleocapsid	[59]
A24:02	VYIGDPAQL VYFLQSINF QYIKWPWYI	ORF1ab ORF3a Spike	[59]
B07:02	IPRRNVATL RPDTRYVL SPRWYFYYL	ORF1ab ORF1ab Nucleocapsid	[59]

**Table 2 vaccines-11-00548-t002:** Identification of presented SARS-CoV-2 HLA-I peptides using HLA peptidomics.

HLA-I Allele	Peptide of SARS-CoV-2	Parent Protein	Ref.
A02:01	GPMVLRGLIT GLITLSYHL MLLGSMLYM LEDKAFQL DEFVVVTV SLEDKAFQL KAFQLTPIAV ELPDEFVVV ELPDEFVVVTV	Out-of-frame S.iORF1/2 (also known as ORF2b) Out-of-frame ORF9b	[18]
A02:01	YLNSTNVTI STSAFVETV FGDDTVIEV FASEAARVV	nsp3 nsp2 nsp3 nsp2	[18]
A24:02, A02:05, A68:01	APHGHVMVEL EIKESVQTF LATNNLVVM EEFEPSTQYEY SEFSSLPSY FAVDAAKAY KRVDWTIEY VATSRTLSY IRQEEVQEL APRITFGGP EILDITPCSF EILDITPCSFG HADQLTPTW KNIDGYFKIY NATNVVIKV QLTPTWRVY VGYLQPRTF	nsp1 nsp2 nsp2 nsp3 nsp8 nsp10 nsp14 Membrane ORF7a Nucleocapside Spike Spike Spike Spike Spike Spike Spike	[18]
B07:02 A02:01 and C07:02	GPMVLRGLIT SLEDKAFQL	ORF S.iORF1/2 ORF9b	[36]
B40:01 A68:01 A68:01 A24:02 A68:01 C15:02 B07:02 A03:01 A68:01 A68:01	NEVAKNLNESL TGSNVFQTR STTTNIVTR YYTSNPTTF FTIGTVTLK HSSGVTREL APRITFGGP RITFGGPSD NAPRITFGGP ITFGGPSDSTGSNQNGER	Spike Spike nsp3 nsp3 ORF3a nsp1 Nucleocapside Nucleocapside Nucleocapside Nucleocapside	[36]

**Table 3 vaccines-11-00548-t003:** Identification of presented SARS-CoV-2 HLA-II peptides using HLA peptidomics.

HLA-II Allele	Peptide of SARS-CoV-2	Parent Protein	Ref.
A11:01	KDGIIWVATEGALN	Nucleocapsid	[36]
DRB1*01:02	VYSRVKNLNSSRVPD	Envelope	[36]
DRB1*01:02	YYKLGASQRVAGDS SYYKLGASQRVAGDSG SYYKLGASQRVAGDS SYYKLGASQRVAG SYYKLGASQRVA LSYYKLGASQRVAGDSG LSYYKLGASQRVAGDS LSYYKLGASQRVAGD LSYYKLGASQRVAG LSYYKLGASQRVA TLSYYKLGASQRVAGDSG TLSYYKLGASQRVAGDS TLSYYKLGASQRVAGD TLSYYKLGASQRVAG TLSYYKLGASQRVA RTLSYYKLGASQRVAGDSG RTLSYYKLGASQRVAGDS RTLSYYKLGASQRVAGD RTLSYYKLGASQRVAG RTLSYYKLGASQRVA	Membrane	[36]
DRB1*01:02	TKAYNVTQAFGRRGPE TKAYNVTQAFGRRGP ATKAYNVTQAFGRRGPE ATKAYNVTQAFGRRGP TATKAYNVTQAFGRRGPEQ TATKAYNVTQAFGRRGPE TATKAYNVTQAFGRRGP KPRQKRTATKAYNVTQAF KPRQKRTATKAYNVTQA	Nucleocapsid	[36]
DRB1*01:02	RDISTEIYQAGSTPCNGVEG DISTEIYQAGSTPCNG DISTEIYQAGSTPC ISTEIYQAGSTPCNG	RBM located within the RBD of spike	[37]
B1*03:01 B1*04:01 B1*07:01 B1*07:01 B3*02:02 B4*01:03 B1*04:01 B1*15:01 B3*02:02 B1*04:01 B1*07:01 B1*04:01 B3*01:01	FTRGVYYPDKVFRS SFTRGVYYPDKVFRSSVLHS PPAYTNSFTRGVYYPD SSVLHSTQDLFLPF TRGVYYPDKVFRSSVLH LLPLVSSQCVNLTT LHSTQDLFLPFFSN LHSTQDLFLPFFSNVT TTLDSKTQSLLIVNNATNVVIKV LGVYYHKNNKSW NIDGYFKIYSKHTPINLVRD SETKCTLKSFTVEKGIYQTS TGTGVLTESNKKFLPFQQFGRDIA	Spike	[37]

## Data Availability

Data are contained within the article.

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
