# Peer review of "HLA-I and HLA-II Peptidomes of SARS-CoV-2: A Review"

_vaccines, 2023, doi:10.3390/vaccines11030548_

Round 1
Reviewer 1 Report
Opinion on the manuscript „HLA-I and HLA-II peptidome of SARS-CoV-2: A review” by Abd El-Baky, Amara and Redwan.
In this review, the Authors discuss the immunology of SARS-CoV-2 infection and HLA class I (HLA-I) and class II (HLA-II) immunopeptidomics based on bioinformatics and on mass spectrometry. The text is illustrated by informative figures and by tables listing HLA-I- and HLA-II-presented peptides identified using both methods mentioned above. The review is based on 83 references, mostly from years 2020-2021 (but only two from 2022), as the SARS-CoV-2 infection and COVID-19 are very recent.
The text is clear and written in good English. The Authors focus their attention on prediction of SARS-CoV-2 immunopeptidome based on protein sequences and HLA-I and HLA-II binding preferences. However, they miss one more important factor contributing to the HLA-I immunopeptidome: the trimming of peptides, coming from cytosol to endoplasmic reticulum, by endoplasmic reticulum aminopeptidases ERAP1 and ERAP2. These two enzymes may either shorten peptides too long to be bound by HLA-I and thus produce epitopes fitting HLA-I peptide-binding groove, or, in contrast, trim too much, to the length too short to be bound. Depending on polymorphic ERAP variants, this trimming may be more or less efficient, influencing peptide repertoire bound by a given HLA-I allotype [Lopez de Castro, Front. Immunol. (2018) 9:2463. doi: 10.3389/fimmu.2018.02463]. Different ERAP1 and ERAP2 allotypes have different levels of expression, activity and substrate specificity, and are inherited in different combinations [Kuiper et al., Human Molecular Genetics, 2018, 27:4333–4343, doi: 10.1093/hmg/ddy319]. Thus, the immunopeptidome depends not only on extreme polymorphism of HLA-I, but also on much lower, but still important polymorphism of ERAPs. Aminopeptidase trimming may be a significant filter in determining which peptides can be presented by HLA-I. I think this information does not exceed the topic of this review and may rather enrich it. The more so that ERAPs have been described as a factor influencing the immune response to SARS-CoV-2 [Stamatakis et al. J. Proteome Res. 2020, 19, 4398−4406, doi.org/10.1021/acs.jproteome.0c00457; Saulle et al. Human Immunology 2021, 82:551–560, doi.org/10.1016/j.humimm.2021.05.003].
In conclusion, the review by Abd El-Baky, Amara and Redwan provides big body of interesting informations. After supplementing with the role of ERAPs it should be acceptable for publication.
Author Response
Comment: In conclusion, the review by Abd El-Baky, Amara and Redwan provides big body of interesting information. After supplementing with the role of ERAPs it should be acceptable for publication.
Response: Appreciate your suggestion for improvement of the manuscript by the role of ERAPs and modifying the text accordingly. We surveyed the suggested references, two of them added to the manuscript along with three new ones, and excluded the following references as their scope is different from SARS-Cov-2:
[1] López de Castro JA (2018) How ERAP1 and ERAP2 Shape the
Peptidomes of Disease-Associated MHC-I Proteins. Front. Immunol. 9:2463.
doi: 10.3389/fimmu.2018.02463. This review focus on the effects of ERAP1 and ERAP2 polymorphism and expression on shaping the peptidome of four diseases associated with MHC-I molecules: birdshot chorioretinopathy (HLA-A∗29:02), ankylosing spondylitis (HLA-B∗27), Behçet’s disease (HLA-B∗51), and psoriasis (HLA-C∗06:02).
[2] Kuiper JJW, et al., Functionally distinct ERAP1 and ERAP2 are a hallmark of HLA-A29-(Birdshot) Uveitis. Hum Mol Genet. 2018;27(24):4333-4343. This research studied effect of ERAP1 and ERAP2 on Birdshot Uveitis (Birdshot), a rare eye condition that affects HLA-A29-positive individuals.
Reviewer 2 Report
The manuscript “ HLA-I and HLA-II peptidome of SARS-CoV-2: A review” addresses the 23 detection of SARS-CoV-2 viral epitopes on HLA-I and HLA-II using bioinformatics prediction and 24 mass spectrometry (HLA peptidomics). The abstract has an introduction from line 10 to 23 and then states purpose. The manuscript will benefit if the intro I lines 10 to 22 is cut in half and more data on the peptidome is described. Talk about which viral protein have more peptides mapped and why it is important for immune response in abstract.
Like wise lines 30 to 80 is repetitive of wha tis know and introduction to the topic starts on line 82. Lines 30 to 80 can also be trimmed in half.
Figure 1 is excellent and reduces the extensive review of cell mediated immunity in section 2 which also needs to be reduced.
Section 3 on immunopeptidomics really is the start on new analysis for this paper
The conclusion is very short. There is not much context discussion on the tabulation of all the peptide fragments
Author Response
Comment 1: The abstract has an introduction from lines 10 to 23 and then states the purpose. The manuscript will benefit if the intro I lines 10 to 22 is cut in half and more data on the peptidome is described. Talk about which viral protein have more peptides mapped and why it is important for immune response in the abstract.
Response: Lines from 10 to 22 in abstract were trimmed and the suggested addition was done.
Comment 2: Likewise, lines 30 to 80 is repetitive of what is known and the introduction to the topic starts on line 82. Lines 30 to 80 can also be trimmed in half.
Response: The suggested lines were trimmed as possible for readability of the introduction.
Comment 3: Figure 1 is excellent and reduces the extensive review of cell mediated immunity in section 2 which also needs to be reduced.
Response: Reduced as suggested.
Comment 4: The conclusion is very short. There is not much context discussion on the tabulation of all the peptide fragments.
Response: Conclusion was developed as suggested.
Round 2
Reviewer 2 Report
thanks for the reductions and additions
Manuscript is improved